# Health Literacy: Current Status and Challenges in the Work of Family Doctors in Bosnia and Herzegovina

**DOI:** 10.3390/ijerph16081324

**Published:** 2019-04-12

**Authors:** Nevena Todorovic, Aleksandra Jovic-Vranes, Bosiljka Djikanovic, Natasa Pilipovic-Broceta, Nadja Vasiljevic, Maja Racic

**Affiliations:** 1Family Medicine Department, Faculty of Medicine, University of Banja Luka, Save Mrkalja 14, 78 000 Banja Luka, Bosnia and Herzegovina; natasa.pilipovic.broceta@gmail.com; 2Institute of Social Medicine, Faculty of Medicine, University of Belgrade, Dr. Subotica 8, 11000 Belgrade, Serbia; aljvranes@yahoo.co.uk (A.J.-V.); bosiljka_djikanovic@yahoo.com (B.D.); 3Institute of Hygiene and Medical Ecology, Faculty of Medicine, University of Belgrade, Dr. Subotica 8, 11000 Belgrade, Serbia; nvas@carlosoft.net; 4Department for Primary Health Care, Faculty of Medicine, University of East Sarajevo, Studentska 5, 73300 Foca, Bosnia and Herzegovina; porodicnamedicina@gmail.com

**Keywords:** health literacy, S-TOFHLA, primary health care, Bosnia and Herzegovina

## Abstract

Health literacy (HL) has become an important area of research. The aim of this study was to evaluate the HL of primary healthcare patients in the Republic of Srpska (RS), Bosnia and Herzegovina (B&H) and to identify socioeconomic and health factors associated with HL. This cross-sectional study among 768 patients was conducted in two healthcare centres between March and May 2017, using the Short Test of Functional Health Literacy in Adults (S-TOFHLA). Analysis was done using descriptive and inferential statistics (a chi-squared test and logistic regression). Inadequate and marginal HL was found in 34,6% of respondents. Socioeconomic and self-reported health factors were significantly related to HL. An age of 55 years and over (OR 1.02), living in a rural environment (OR 2.25), being divorced (OR 3.32), being insufficiently physically active (OR 1.29), having poor income (OR 1.96), having more than three chronic diseases (OR 1.94), and poor health (OR 1.59) were significantly corelated with inadequate and marginal HL. The results of our study indicate that a low level of HL is related to the elderly, having a divorce, having a rural residence, poor income, having more than three chronic diseases, poor health, and insufficient physical activity. Further evaluation, monitoring, and activities to improve HL are of great importance for patients’ health outcomes.

## 1. Introduction

Health literacy (HL) is one of the most important pillars of health promotion and disease prevention [1]. It is defined as knowledge, motivation, and competency to approach, understand, evaluate, and apply the information necessary for making healthcare decisions to maintain good health and quality of life [2,3]. Inadequate health literacy is more common in people who are older (especially older than 65 years), people who have low levels of education, poor health, and more chronic diseases, people who are poor, and people who are members of certain racial and ethnic groups, and are refugees and immigrants [4]. The level of health literacy is related to cooperation between patients and physicians, results of treatment, and quality of healthcare [5]. Health care is a mutual responsibility of patients and physicians. They have to communicate in an understandable way in order to make responsible decisions concerning the patient’s health [6]. Daily, family physicians tackle problems caused by inadequate health literacy. These include inappropriate use of medication [7], misunderstanding treatment advice, false self-perception of health, few preventive examinations, frequent physician office visits, poor knowledge of self-help skills, difficulties in communication [5], providing incomplete health histories, and ill preparation for and failure to appear for appointments, diagnostic tests and procedures [8]. Evaluation of health literacy, or literacy in general, cannot be adequately assessed by asking whether the patient has understood the discussion or knows how to read or write [9]. There are many tests to evaluate functional health literacy [10] in a health care system, through evaluation of word recognition, reading tests, and reading comprehension [11]. One of the most common tests and one which is valid in the local language is the Test of Functional Health Literacy in Adults (S-TOFHLA) [9,10,12,13]. Our goal was to use S-TOFHLA to evaluate health literacy of primary healthcare users in the Republic of Srpska (RS), Bosnia and Herzegovina (B&H) and to identify and analyze socioeconomic and health factors linked to the level of health literacy.

## 2. Materials and Methods

### 2.1. Study Design

This cross-sectional study was conducted in two primary healthcare centres (PHC), Prijedor and Bijeljina, between 1 March and 31 May 2017 and included 768 patients (384 respondents in each PHC) registered with 89 family medicine physicians. The study was conducted in accordance with the World Medical Association Declaration of Helsinki (1975, 2013). The ethics committees of the Faculty of Medicine of the University of Belgrade and of both PHCs gave written approval. Every patient included in the study received a written explanation of the goals, methods, and purpose of this research and gave written consent prior to enrolment in this study.

### 2.2. Selection and Description of Participants

The study was conducted by eight investigators (physicians or nurses) trained in the use of S-TOFHLA. Respondents were randomly selected from the patients registered with one of the family practices in the PHC in Prijedor, and then in Bijeljina (north-western and north-eastern part of the RS, B&H). The representative sample of the population in the Republic of Srpska was determined by the sample size calculation. With a population size of 934,725 inhabitants older than 18 years of age in the Republic of Srpska, the proportion of inadequate health literacy of 32% (determined in a previous pilot study) [13], an error margin of 5%, a 95% Confidence Interval (CI) and an expected attrition rate of least 10%, the minimum required number of respondents was found to be 844. Every Wednesday, in the study period, investigators consecutively selected a sample of eight patients visiting their family physicians for any reason, until the desired sample size was achieved. The literacy skills of each prospective study subject were assessed by asking them to read and sign the informed consent form. Individuals unwilling to cooperate, with visual disorders, organic brain damage, mental disorders, alcohol intoxication, and a health profession education background, as well as illiterate persons and patients younger than 18 years were not enrolled in the study.

### 2.3. Data Collection

S-TOFHLA and a sociodemographic questionnaire were administered by investigators after the respondent’s visit to the family physician. Respondents individually gave written answers to questions from both questionnaires. It took approximately 30 min for the following activities: Application of exclusion criteria, explanation of objectives and methods of research, written consent for participation, and completion of both questionnaires.

### 2.4. Instrument

We used the short version of the TOFHLA questionnaire (Short Test of Functional Health Literacy in Adults, S-TOFHLA) [10] and a sociodemographic questionnaire [13]. Translation from the English language and validation of the STOFHLA questionnaire in the Serbian language was conducted previously in the same geographic regions and published elsewhere. It was shown that S-TOFHLA had good internal consistency, reliability, and construct validity compared with the long version—the TOFHLA [12,13].

The sociodemographic questionnaire included 23 items on demographic, social, economic, as well as health characteristics and was administered in the validation study [13]. We measured self-perceived material status and the health characteristics of the respondents encompassing self-assessment of general health, use of health services, the presence of chronic illness, and bad habits. The use of health services was evaluated through the number of visits to family medicine specialists, other specialists in the state and private sector, and the number of hospitalizations over the past 12 months. Respondents were asked to list any chronic diseases and medications taken (used at least seven days due to illness). Respondents were asked about their risk behaviour, which included smoking, alcohol intake, body mass index, and insufficient physical activity. We measured self-perceived life satisfaction [13].

### 2.5. Description of Instrument

The STOFHLA questionnaire is divided into two parts—A and B (which correspond to the sections on the standard version of the questionnaire) [10]. Part A consists of 16 items (carries 16 points) and refers to instructions that the patient receives before performing diagnostic procedures for the upper gastrointestinal tract examination. Part B has 20 items (carries 20 points) and examines the knowledge of patients’ rights and obligations in the healthcare system. The time provided for completing the questionnaire is seven minutes. The respondents receive 1 point for each correct answer and 0 points if they do not answer or if the answer is incorrect. The maximum number of possible points obtained in the shortened version of the questionnaire is 36. We labeled health literacy as inadequate HL (score, 0–16), marginal HL (score, 17–22), and adequate HL (score, 23–36) [10].

### 2.6. Statistical Analysis

Data were entered and analyzed using the Statistical Package for Social Sciences, version 22 (IBM SPSS Statistics 22 Windows, Version 22.0. Armonk, NY: IBM Corp. 2013). The chi-squared test was used to assess the significance of differences by patient characteristics and functional HL categories. Demographic, social, economic, and health characteristics were independent variables, and HL was a dependent variable. Their relationship was assessed by multiple logistic regression. For this analysis, the inadequate and marginal categories were combined into one dependent variable. The analysis of logistic regression was organized in two stages. *p* < 0.05 was considered significant.

## 3. Results

### 3.1. Sociodemographic Characteristics of Respondents

The study included 768 respondents out of 844 who were invited to participate in both PHC centers. Sixty-nine questionnaires were invalid and therefore excluded from the analyses. Seven respondents asked to withdraw from the study as they found the survey time-consuming. Of the 768 participants in the assessment of the health literacy of primary healthcare respondents, 442 (57.5%) were females and 326 (42.5%) were males, with 65.6% living in rural and 34.4% living in urban areas. The average age for all respondents was 49.91 ± 17.45 years. For male respondents, average age was 50.27 ± 17.98 and for females, it was 49.70 ± 17,03 years, but this gender difference was not statistically significant. Forty-four point eight (44.8) percent of respondents were younger than 44 and most lived in urban areas, while 24.3% were older than 65, with most living in rural areas. The education level of respondents indicated that 19.7% were highly educated (12 or more years of education), 60.5% were moderately educated (8–12 years of education), and 19.8% had low levels of education (less than 8 years). Highly educated respondents mostly lived in urban areas, and those with low levels of education mainly lived in rural areas. Marital status data indicated that 57.3% of respondents were married (formally or informally). Thirty-eight percent of respondents who were employed, mostly lived in urban areas, while the unemployed tended to live in rural areas. Fifty-three point five (53.5) percent of respondents had an average financial status, and 17.5% were poor (Table 1).

### 3.2. Health Characteristics of Respondents

Thirty-eight point six (38.6) percent of respondents rated their health as average. Of the 46.5% of respondents with chronic diseases, the average number of diseases was 2.67 (SD = 2.579). Most respondents who reported good health lived in urban areas, and those reporting poor health lived in rural areas. The health status of respondents was statistically significantly correlated to their place of residence (*p* < 0.001). Forty-two point one (42.1) percent of respondents were inadequately physically active, and they mostly had inadequate health literacy. Physical activity of the respondents was significantly related to health literacy (*p* < 0.001) (Table 1).

### 3.3. Health Literacy of Respondents

The distribution of respondents according to their level of health literacy is presented in Figure 1. In the total sample, 24.6% of respondents had inadequate, 10% had marginal, and 65.4% had adequate health literacy. Respondents from Bijeljina had lower levels of health literacy compared to their counterparts in Prijedor (*p* < 0.001).

Inadequate and marginal health literacy was more frequently identified among individuals who were older than 65 years (*p* < 0.001), unmarried (*p* = 0.001), unemployed (*p* = 0.002), and who had poor income (*p* < 0.001), a low education level (*p* < 0.001), or a residence in rural areas (*p* < 0.001). Physically active respondents (*p* < 0.001), having a good health self-perception (*p* < 0.001), good overall quality of life (*p* = 0.017) or fewer hospital treatments/hospitalizations (*p* = 0.007), and visiting their family physician less reported higher levels of health literacy (Table 2).

### 3.4. Determinants of Inadequate and Marginal Health Literacy

Independent factors associated with inadequate and marginal health literacy were older age (OR, 1.02; CI 95%, 1.00–1.05; *p* = 0.03), being divorced (OR, 3.32; CI 95%, 1.26–8.72; *p* = 0.02), residing in a rural area (OR, 2.25; CI 95%, 1.31–3.87; *p* < 0.01), poor income (OR, 1.96; CI, 1.96; 1.09–3.53; *p* = 0.03), multimorbidity (OR, 1.94; CI 95%, 1.02–3.67; *p* = 0.04), poor self-perception of health (OR, 1.59; CI 95%, 1.12–2.26; *p* = 0.04), and insufficient physical activity (OR, 1.29; CI 95%, 1.16–1.45; *p* < 0.01) (Table 3).

## 4. Discussion

### 4.1. Summary of Results

The average value of the S-TOFHLA score for all respondents in the current study was 24.83 (SD = 10.37) out of a total score of 36, showing that the majority of respondents had adequate health literacy. The analysis provided in accordance with the health literacy model and the overall health status of the respondents indicated that people with inadequate or marginal health literacy suffer from a higher number of chronic diseases, are less able to predict their personal health, see their family doctors more frequently, and have more frequent hospital treatments. The highest degree of inadequate health literacy was seen in those who were not married, were retired, were older, lived in rural areas, had lower education levels and poor income, had more than three chronic diseases, suffered poor health, and were insufficiently physically active. This study concluded that these socioeconomic and health characteristics correlate with the overall level of health literacy.

### 4.2. Comparision with International Population-Based Studies

Comparative results of health literacy research conducted in different countries are different to our results. In general, the proportion of individuals with inadequate health literacy in the Republic of Srpska (B&H) was higher compared to the population in developed countries and close to the results from Bulgaria 26.9%, then followed by Serbia 20.0%, Austria 18.2%, Spain 12.4%, Germany 11.0%, Greece 10.3%, and the Netherlands 10.2%. Inadequate health literacy was less represented among the population in Ireland 1.8%, Finland 4.8%, Poland 7.5%, and Hungary 8% [14,15,16,17].

### 4.3. Comparison with the Current Literature

Previous research has shown that low socioeconomic status represents a significant risk factor that influences health literacy, and that deprivation, poverty, and social inequality in the population are profoundly related to inadequate health literacy [3,14]. People with a lower socioeconomic status (unemployment, poor income) suffer from poor health, are more prone to chronic illnesses and injury, utilize preventative services less, exhibit worse mental health, have a higher mortality rate, etc. [15,16,17,18]. In our study, 34.6% of respondents demonstrated inadequate and marginal health literacy, a figure similar to other studies where the prevalence of inadequate health literacy varied between 32% and 59% [8,12,13,14,19,20]. Additionally, the healthcare of the respondents was not significantly influenced by gender [13,21]; however, other studies [12,22,23,24] that referenced gender found it was a relevant factor in determining health literacy. The overall health literacy score of study respondents decreased as their age increased, a similar pattern to many other studies [11,12,13,14,15,16,17,18,19,20,21,22,24,25,26,27,28,29,30], while research groups in Japan and Switzerland did not categorize age as a relevant factor in individual health literacy in their study populations [31,32]. We found that health literacy profoundly decreased in respondents that were not in a marriage relationship, a relationship similar to that found in a study in the United States [33], but not in an Italian study [19]. In this study, health literacy among respondents living in urban communities was significantly higher in comparison to those who lived in rural communities, as was found by Parker [34]. Education level also had a significant impact on health literacy in several other studies [12,13,14,17,21,29,30,35], but not one in Japan [31].

Our research concluded that the best S-TOFHLA score was achieved by highly-educated respondents, while the poorest scores belonged to respondents with low levels of education, as demonstrated by other academic research conducted in this area [12,13,14,17,18,22,23,24,25,26,27]. Full-time employed respondents in our study were most likely to be classified as adequately health literate, as has been seen in other studies [13,18,27,29,31]. The overall health literacy score achieved by our respondents varied in terms of their material status/income status, from higher scores in those living with good income to poorer health care literacy scores in respondents with poor income; this is also in agreement with other studies [29,35,36].

In a majority of examples, respondents who described their health as decent/solid had adequate health literacy. On the other hand, more than half of inadequately or marginally health literate respondents were affected by one or more diseases. The total number of family doctor visits and number of hospitalizations/hospital treatments was profoundly associated with the health-care literacy score, as was the case in other studies [29,30,37]. The respondent’s total number of diseases was strongly and inversely associated with their health literacy level. The correlation of having three or more diseases in 50% of respondents with adequate health literacy is similar to the findings of other studies [12,13,14,17,18,24,25,26,27,32]. In line with previous studies, the overall health literacy scores of our respondents decreased with the degradation of an individual’s health status. Respondents with a recognized better health status are more often classified within the category of having adequate health literacy, while the respondents suffering from poor health status are in most cases classified as having inadequate health literacy. Lifestyle choices may explain some of these links. Individuals with adequate literacy are more likely to consume healthy foods and have regular physical activity [30,31,32]. Health literacy level may also influence risk as well as disease perception and their relation to healthy behavior, ultimately leading to the prevention of non-communicable diseases and multimorbidity [38,39,40]. However, the evidence is inconclusive and merits further research.

### 4.4. The Significance for Clinical Practice

This research indicates that inadequate health literacy is common among vulnerable population groups and must be recognized by family doctors caring for these patients. Family doctors are impacted by, should expect and should address health literacy problems in patients who suffer multiple diseases, fail to take advantage of preventive and health promotion services, and who do not comply well with treatment [13].

### 4.5. The Significance for Future Analysis

This study is the assessment of health literacy levels and their relationship to socioeconomic variables, health self-assessment, the presence of chronic illnesses, and the material/income status majority of patients receiving primary healthcare in RS and B&H [13]. These results should inform policymakers of the need to consider and address health literacy in policy making and enhance the knowledge and skills of the medical staff with regard to the health literacy of patients seen in medical and healthcare institutions. It is imperative to integrate health literacy data into the clinical and public healthcare practice and long-term research programs and health literacy initiatives in order to increase the health literacy of the population and address its impact on the providers of healthcare services [13].

### 4.6. Study Limitations

The sample of this study was randomly selected but only reflects health literacy in two cities in RS. The study had a limited sample size. It should be replicated with a larger, more representative population of the nation using available data from all healthcare centers in order to provide a more precise analysis of overall health literacy. The study does not apply to those unwilling to cooperate: The illiterate, those who did not speak the official languages of RS (B&H), those with insufficient eyesight to read the S–TOFHLA questionnaire, those with organic brain damage, the mentally ill, those who were incoherent or were having a psychotic episode, the alcohol intoxicated, and those with a medical, dental, or pharmacologic education. The limitation of this study may also be the potential bias of patient selection. We assume that the inadequate HL level could be a reason for a refusal to participate in the study. Also, study participants were only individuals who have visited their family physicians and therefore might have had different health status and level of health literacy in comparison to those who are rarely exposed to medical encounters. Furthermore, despite the strong recommendations and detailed information provided by the authors of the study, we cannot rule out the choice of patients based on the preferences of investigators. In addition, the study design is cross-sectional, so a causal relationship was not analyzed.

## 5. Conclusions

More than a third of our respondents had inadequate individual health literacy. Socioeconomic and demographic characteristics strongly impacted health literacy. Inadequate health literacy was related to socioeconomic and health characteristics including age, divorce, rural residence, poor income, more than three chronic diseases, poor health, and insufficient physical activity.

## Figures and Tables

**Figure 1 ijerph-16-01324-f001:**
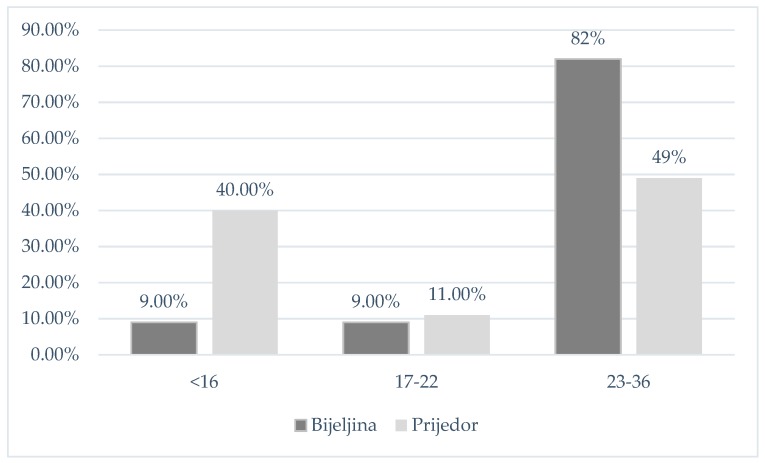
Levels of health literacy by town.

**Table 1 ijerph-16-01324-t001:** Socio-demographic characteristics of respondents.

	N	%
Gender		
Male	326	(42.5)
Female	442	(57.5)
Age, years		
Below 44	298	(38.8)
45–54	132	(17.2)
55–64	151	(19.7)
65 and above	186	(24.3)
Marital status		
Married	437	(57.3)
Other	325	(42.7)
Low: Primary school or less (≤8)	151	(19.8)
Moderate: Secondary school (8–12)	462	(60.5)
High: High and higher school (>12)	150	(19.7)
Area		
Urban	264	(34.4)
Rural	503	(65.6)
Employment		
Employed	294	(38.3)
Other	474	(61.7)
Income (per person)		
Poor (<700 BAM *)	134	(17.5)
Average (700–900 BAM)	410	(53.5)
Good (>900 BAM)	223	(29.1)
Insufficient physical activity risk		
No	323	(42.1)
Yes	444	(57.9)
Chronic diseases		
Without	410	(53.5)
One	156	(20.4)
Two	74	(9.7)
Three	49	(6.4)
More than three	77	(10.1)
Self-perception of health		
Poor	86	(11.3)
Average	294	(38.6)
Good	382	(50.1)
Life satisfaction		
Low	165	(21.5)
Average	390	(50.9)
High	211	(27.5)
Municipality		
Bijeljina	384	(50.0)
Prijedor	384	(50.0)
Total	768	

* BAM-Bosnian konvertible mark.

**Table 2 ijerph-16-01324-t002:** Distribution of respondents by health literacy level.

	S-TOFHLA Score Categories						
Characteristics	Inadequate		Marginal		Adequate		*p* Value ^a^
	(n = 189)	%	(n = 77)	%	(n = 502)	%	
Area							<0.001
Urban	36	(19.0)	25	(32.5)	203	(40.5)	
Rural	153	(81.0)	52	(67.5)	298	(59.5)	
Gender							0.244
Male	88	(46.6)	36	(46.8)	202	(40.3)	
Female	101	(53.4)	41	(53.2)	299	(59.7)	
Age, years							<0.001
Below 44	39	(20.7)	25	(32.5)	234	(46.6)	
45–54	18	(9.6)	11	(14.3)	103	(20.5)	
55–64	44	(23.4)	20	(26.0)	87	(17.3)	
65 and above	87	(46.3)	21	(27.3)	78	(15.5)	
Marital status							0.001
Married	86	(46.0)	49	(64.5)	302	(60.5)	
Other	101	(54.0)	27	(35.5)	197	(39.5)	
Employment							0.002
Employed	53	(28.0)	27	(35.1)	214	(42.6)	
Other	136	(72.0)	50	(64.9)	288	(57.4)	
Education (years of school completed)							<0.001
Low: Primary school or less (≤8)	62	(32.8)	20	(26.0)	69	(13.9)	
Moderate: Secondary school (8–12)	111	(58.7)	46	(59.7)	305	(61.4)	
High: High and higher school (>12)	16	(8.5)	11	(14.3)	123	(24.7)	
Income (per person)							<0.001
Poor (<700 BAM)	56	(29.6)	20	(26.0)	58	(11.6)	
Average (700–900 BAM)	96	(50.8)	38	(49.4)	276	(55.1)	
Good (>900 BAM)	37	(19.6)	19	(24.7)	167	(33.3)	
Self-perception of health							<0.001
Poor	66	(35.3)	26	(34.7)	32	(6.4)	
Average	75	(40.1)	41	(54.7)	178	(35.6)	
Good	46	(24.6)	8	(10.7)	290	(58.0)	
Chronic diseases							<0.001
Without	86	(45.5)	37	(48.1)	287	(57.4)	
One	40	(21.2)	17	(22.1)	99	(19.8)	
Two	16	(8.5)	6	(7.8)	52	(10.4)	
Three	10	(5.3)	6	(7.8)	33	(6.6)	
More than three	37	(19.6)	11	(14.3)	29	(5.8)	
Insufficient physical activity risk							<0.001
No	39	(20.6)	27	(35.1)	257	(51.3)	
Yes	150	(79.4)	50	(64.9)	244	(48.7)	
Family doctor visits							0.017
No visits	37	(20.4)	19	(10.5)	125	(69.1)	
1–2 visits	63	(27.5)	27	(11.8)	139	(60.7)	
3–4 visits	41	(30.8)	8	(6.0)	84	(63.2)	
5–10 visits	23	(22.3)	4	(3.9)	76	(73.8)	
More than 10	24	(20.0)	19	(15.8)	77	(64,2)	
Number of hospitalisations							0.007
No hospitalisations	145	(77.1)	67	(87.0)	438	(87.8)	
1–2 times	37	(19.7)	7	(9.1)	50	(10.0)	
More than 2	6	(3.2)	3	(3.9)	11	(2.2)	
Change of health							<0.001
Better	30	(16.3)	18	(24.0)	147	(29.6)	
Same	96	(52.2)	45	(60.0)	279	(56.1)	
Worse	58	(31.5)	12	(16.0)	71	(14.3)	
Municipality							<0.001
Bijeljina	34	(8.8)	36	(9.4)	314	(81.8)	
Prijedor	155	(40.4)	41	(10.7)	188	(48.9)	

Data are given as number (%). ^(**a**)^ According to the chi-squared test.

**Table 3 ijerph-16-01324-t003:** Factors associated with health illiteracy according to S-TOFHLA score (adequate versus marginal and inadequate)—multivariate logistic regression analysis.

Variable	Univariate	Multivariate
	OR	95% CI	*p*	OR	95% CI	*p*
Age, years	1.04	1.03	‒	1.05	0.000	1.02	1.00	‒	1.05	0.03
Marital status, divorced	4.2	2.43	‒	7.28	0.000	3.32	1.26	‒	8.72	0.02
Place of residence, rural	2.29	1.63	‒	3.21	0.000	2.25	1.31	‒	3.87	0.00
Income, low	3.06	2.09	‒	4.47	0.000	1.96	1.09	‒	3.53	0.03
Number of chronic diseases, more than three	3.58	2.19	‒	5.83	0.000	1.94	1.02	‒	3.67	0.04
Taking medicines	0.53	0.32	‒	0.88	0.015					
Self-perception of health	1.96	1.64	‒	2.35	0.000	1.59	1.12	‒	2.26	0.01
Physical activity, insufficient	1.41	1.3	‒	1.52	0.000	1.29	1.16	‒	1.45	0.00
Tertiary education	0.34	0.22	‒	0.54	0.000					
Employment status, retired	3.16	2.19	‒	4.55	0.000					
Frequency of hospitalization	1.54	1.12	‒	2.13	0.009

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
