# Peer review of "Health Literacy: Current Status and Challenges in the Work of Family Doctors in Bosnia and Herzegovina"

_ijerph, 2019, doi:10.3390/ijerph16081324_

Round 1

Reviewer 1 Report

This study has been carried out well and shows an extensive collection of data about health literacy of primary care participants. The study is an update and presents more extensive data than the previous publication by the same authors.

On which hypothesis is the sample size calculation based? Presumably finding an accurate assessment of health literacy that is representative of Bosnia Herzegovina (S-TOFHLA) or only Republic of Srpska? (line 79). Assuming that this sample size was large enough to generate an accurate assessment of functional health literacy, what does this finding mean in comparison with other countries in European region, for example? While this is discussed qualitatively in the discussion, maybe cross country comparisons of average functional health literacy could be shown?

How many potential participants were excluded due to mental illhealth or neurodegeneration? As illiteracy is also an exclusion criterion – how was this assessed and how many people were excluded on this basis. Was ethnicity or first language of participants recorded and why is this not shown?

The demographic data table (table 1) should not have health literacy results as well. These should be presented in table 2. In fact, it may be clearer to present the percentage findings of inadequate, marginal and adequate as a bar chart rather than numerically.

The first finding I would like to see is the frequency distribution of S-TOFHLA scores. Maybe showing the two geographic areas separately and maybe also show distributions for the two parts of the questionnaire. Furthermore it would be informative to see some more data about the two parts of the questionnaire – do these parts have equal weighting in the overall score? For example only 1.6% of participants answered the second part ‘correctly’ – yet presumably many of these were judged to have ‘adequate’ health literacy. Line 156 & paragraph appears conflicting – initial statement is 64% have adequate health literacy, then the next statement is that a minority (44.5) answered the easy question correctly. Later in the paragraph are statements about inadequate health literacy – please define this – does it include marginal as well as inadequate categories?

Please clarify line 173-176. Several different characteristics are described and percentages add up to greater than 100, but it is not clear to which categories percentages relate. In places terms such as ‘sufficient health literacy’ and ‘health illiterate’ are used – I would suggest not using the term illiterate – but how do these terms relate to STOFHLA score and the 3 categories defined earlier in the manuscript?

This section, line 173 to 190 is generally confusing – being a list of categories and percentages where sometimes the percentage relates to the independent variable and sometimes the percentage is give of the dependent variable.

Line 195 – my understanding of section 3.4 is that this analysis relates to this phrase within the methods (line 123) “For this analysis, the inadequate and marginal categories were combined into one denoting limited functional health literacy.” If so, “inadequate health literacy” here means “marginal and inadequate” (as in Table 3 legend) and therefore contradicts the earlier definition of “inadequate HL (score, 0-16)”(line 115). Throughout the findings section these definitions of inadequate etc are unclear and seem to change.

Likewise there are variable use of terms such as ‘health literacy’, as well as ‘healthcare literacy’ – are these synonymous?

Line 236 mentions migration as a significant factor but I cannot find data relating to this?

Relationship between low health literacy and multimorbidity and high service use is observed, but not really interpreted. Presumably the authors consider causal relationship and this would be interesting discussion point.

There are many typographical errors, and I’m happy to highlight these in the next draft manuscript.

Author Response

Dear Editor,

Thank you for the effort and time you and reviewers have invested in reviewing the Manuscript " Health literacy: current status and challenges in the work of family doctors in the Bosnia and Herzegovina". All of your suggestions have been discussed with the authors of the work and we have jointly accepted your comments.

Thank you for helping us to improve the quality of the manuscript. All changes are colored red.

Yours sincerely,

 Nevena Todorovic, MD, MsD

In brief, we provide you with explanatory comments:

Reviewer 1#

This study has been carried out well and shows an extensive collection of data about health literacy of primary care participants. The study is an update and presents more extensive data than the previous publication by the same authors.

Comment 1 :On which hypothesis is the sample size calculation based? Presumably finding an accurate assessment of health literacy that is representative of Bosnia Herzegovina (S-TOFHLA) or only Republic of Srpska? (line 79).

Assuming that this sample size was large enough to generate an accurate assessment of functional health literacy, what does this finding mean in comparison with other countries in European region, for example? While this is discussed qualitatively in the discussion, maybe cross country comparisons of average functional health literacy could be shown?

Answer: Answer: Pilot study was previously conducted and the expected inadequate literacy determined. We added this information in methodology section to easy understanding (Lines 79-88). The sample is representative of Republic of Srpska, but could be representative of BiH as all ethnicities were included. The discussion is an additional part that describes the results of international studies on health literacy about different populations.

Comment 2: How many potential participants were excluded due to mental illhealth or neurodegeneration? As illiteracy is also an exclusion criterion – how was this assessed and how many people were excluded on this basis. Was ethnicity or first language of participants recorded and why is this not shown?

Answer: The patients with mental disorders were not enrolled with the study. The participants were selected during their visit to their family physician. If they had one of exclusion criteria, they were skipped. Those who did not have any of the exclusion criteria described in manuscript were asked to read inform consent. If they understood it and were able to sign it, were considered literate. During pilot study, many respondents did not feel comfortable to record their nationality and in main study we decided to discard this information. After the civil war ethnicity questions are a bit difficult to deal and very vulnerable area for many people. Serbian language is one of three official languages spoken in Bosnia and Herzegovina (in addition to Croatian and Bosnian). These languages are the same (are similar), but are called differently due to political reasons (Line 85, 86).

Comment 3: The demographic data table (table 1) should not have health literacy results as well.

Answer: Health literacy results are removed (Table 1)

Comment 4: These should be presented in table 2. In fact, it may be clearer to present the percentage findings of inadequate, marginal and adequate as a bar chart rather than numerically.

Answer: Percentages are presented in Figure 1 as suggested.

Comment 5: The first finding I would like to see is the frequency distribution of S-TOFHLA scores. Maybe showing the two geographic areas separately and maybe also show distributions for the two parts of the questionnaire. Furthermore it would be informative to see some more data about the two parts of the questionnaire – do these parts have equal weighting in the overall score? For example only 1.6% of participants answered the second part ‘correctly’ – yet presumably many of these were judged to have ‘adequate’ health literacy. Line 156 & paragraph appears conflicting – initial statement is 64% have adequate health literacy, then the next statement is that a minority (44.5) answered the easy question correctly. Later in the paragraph are statements about inadequate health literacy – please define this – does it include marginal as well as inadequate categories?

Answer: Detailed explanation of the questionnaire in Serbian language is published in another paper (reference number 13). The authors in this paper did not repeat detailed explanations to avoid autoplarijarism. However, we added to the methods short description of grading.  Parts have equal weighing in the overall score. We removed the sentence related to the second part of questionnaire to avoid readers’ confusion. Where the results were related to inadequate HL we left only the word “inadequate”. If the results were related to both inadequate and marginal, we used “inadequate and marginal”.

Frequency distribution of S-TOFHLA scores by towns is presented in Figure 1.

Comment 6: Please clarify line 173-176. Several different characteristics are described and percentages add up to greater than 100, but it is not clear to which categories percentages relate. In places terms such as ‘sufficient health literacy’ and ‘health illiterate’ are used – I would suggest not using the term illiterate – but how do these terms relate to STOFHLA score and the 3 categories defined earlier in the manuscript?

Answer: Wording has been changed to improve the understanding. The categories of health literacy have been used uniformly in the manuscript as suggested.

Comment 7: This section, line 173 to 190 is generally confusing – being a list of categories and percentages where sometimes the percentage relates to the independent variable and sometimes the percentage is give of the dependent variable.

Answer: Wording has been changed to improve the understanding. The categories of health literacy have been used uniformly in the manuscript as suggested.

Comment 8: Line 195 – my understanding of section 3.4 is that this analysis relates to this phrase within the methods (line 123) “For this analysis, the inadequate and marginal categories were combined into one denoting limited functional health literacy.” If so, “inadequate health literacy” here means “marginal and inadequate” (as in Table 3 legend) and therefore contradicts the earlier definition of “inadequate HL (score, 0-16)”(line 115). Throughout the findings section these definitions of inadequate etc are unclear and seem to change.

Answer: Thank you for your comment.

We made corrections in section “statistical analysis” and “section 3.4”.

Comment 9: Likewise there are variable use of terms such as ‘health literacy’, as well as ‘healthcare literacy’ – are these synonymous?

Answer : Yes. We changed to health literacy uniformly

Comment 10: Line 236 mentions migration as a significant factor but I cannot find data relating to this?

Answer: Sentence has been removed

Comment 11: Relationship between low health literacy and multimorbidity and high service use is observed, but not really interpreted. Presumably the authors consider causal relationship and this would be interesting discussion point.

Answer: Discussion point has been added

Comment 12: There are many typographical errors, and I’m happy to highlight these in the next draft manuscript.

Answer: We reviewed the English language of the entire manuscript and corrected it.

Reviewer 2 Report

Thank you for the opportunity to review this manuscript. In general the manuscript is well written and referenced, and the discussion section includes comparison of study results from previous literature. 

The following are my major comments/ suggestions:

The English language of the entire manuscript should be proofread carefully again to ensure gramatically appropriate before potential publication by the Journal.

The authors classified the participants' incomes as "poor, average, or good", and "low or high", and "bad or worse" throughout the manuscript. I would suggest if possible specify numerical values for the income status for easy comprehension.

What language version of the STOFHLA was used in the study? Some items in the English version of STOFHLA contains items related to American insurance plan with which the respondents in your country may not be familiar. Would these items pose a threat to the validity of the current study result? 

Only 768 out of 844 patients participated in the study. What were the specific reasons those 76 patients not participating in the study, and how was this expected to afffect the study results (e.g. the type of study bias)?

Author Response

Dear Editor,

Thank you for the effort and time you and reviewers have invested in reviewing the Manuscript " Health literacy: current status and challenges in the work of family doctors in the Bosnia and Herzegovina". All of your suggestions have been discussed with the authors of the work and we have jointly accepted your comments.

Thank you for helping us to improve the quality of the manuscript. All changes are colored red.

Yours sincerely,

Mr. sc dr. Nevena Todorovic

In brief, we provide you with explanatory comments:

Reviewer 2#

Thank you for the opportunity to review this manuscript. In general the manuscript is well written and referenced, and the discussion section includes comparison of study results from previous literature. 

The following are my major comments/ suggestions:

Comments 1. The English language of the entire manuscript should be proofread carefully again to ensure gramatically appropriate before potential publication by the Journal.

We reviewed the English language of the entire manuscript and corrected it.

Comments  2. The authors classified the participants' incomes as "poor, average, or good", and "low or high", and "bad or worse" throughout the manuscript. I would suggest if possible specify numerical values for the income status for easy comprehension.

Answer: Thank you for your comments. We tried to use same wording throughout the manuscript. We added numerical values for the income status.

Comments  3. What language version of the STOFHLA was used in the study? Some items in the English version of STOFHLA contains items related to American insurance plan with which the respondents in your country may not be familiar. Would these items pose a threat to the validity of the current study result? 

Answer. Process of translation and validation of STOFHLA in Serbian language is published in another paper (reference number 13). In the previous  study, it was shown that STOFHLA had good internal consistency, reliability , and construct validity compared with the long version of the TOFHLA is published in another paper (reference number 12). The standard version of the TOFHLA questionnaire also contains the third part of C, in which it is more represented by the American insurance plan. We used English version of STOFHLA without items related to American insurance plan.

Comment 4. Only 768 out of 844 patients participated in the study. What were the specific reasons those 76 patients not participating in the study, and how was this expected to afffect the study results (e.g. the type of study bias)?

Answer. Specific reasons for exclusion were described in results section. We also revised sample size calculation paragraph. Patients with predefined exclusion criteria were not enrolled in the study. The investigators consecutively selected a sample of 8 patients from family practices, until the desires sample size was achieved. The sample size of 844 also included expected attrition rate, which was 10%

Round 2

Reviewer 1 Report

The authors have made many improvements and responded to critical points.

The details of recruitment remain unclear and this precludes interpretation of the findings with respect to population or risk of bias. The authors state:

“Every Wednesday, in the study period, investigators consecutively selected a sample of 8 patients from family practices, until the desired sample size was achieved.

Does this indicate that only patients who attended GP (for health concerns) were recruited, or was a recruitment invitation sent to these 8 people who subsequently attended their GP for the purpose of the research survey? If the former, the sampling frame are people who attend the GP and may not relate to people who do not attend their GP - ie may be more healthy, or more health literate. If the latter, it indicates a commitment to the research study, which may bias in favour of higher health literacy. This aspect of method should be clarified.

Author Response

Dear Editor,

Please find enclosed the revised manuscript titled " Health literacy: current status and challenges in the work of family doctors in the Bosnia and Herzegovina" that was corrected, hopefully to the full satisfaction of the editor and reviewers.

We would like to thank the reviewers for careful reading of this manuscript and for constructive suggestions, which help to improve the quality of the article.

The response follows

Reviewer #1:

1.Reviewer’s query. The details of recruitment remain unclear and this precludes interpretation of the findings with

respect to population or risk of bias. The authors state:

“Every Wednesday, in the study period, investigators consecutively selected a sample of 8

patients from family practices, until the desired sample size was achieved.

Does this indicate that only patients who attended GP (for health concerns) were recruited, or was a recruitment invitation sent to these 8 people who subsequently attended their GP for the purpose of the research survey? If the former, the sampling frame are people who attend the GP and may not relate to people who do not attend their GP - ie may be more healthy, or more health literate. If the latter, it indicates a commitment to the research study, which may bias in favour of higher health literacy. This aspect of method should be clarified.

Thank you for your comment.  We clarified method as suggested (Line 84-85). The limitation of this model was added into section Study limitation (Line 286-288)

The corrections are colored yellow.

We would be glad to respond to any further questions and comments that you may have.

Nevena Todorovic